# Digital Traceability in Agri-Food Supply Chains: A Comparative Analysis of OECD Member Countries

**DOI:** 10.3390/foods13071075

**Published:** 2024-04-01

**Authors:** Sylvain Charlebois, Noor Latif, Ibrahim Ilahi, Bibhuti Sarker, Janet Music, Janele Vezeau

**Affiliations:** 1Agri-Food Analytics Lab, Dalhousie University, Halifax, NS B3H 4R2, Canada; jmusic@dal.ca; 2Faculty of Arts and Science, University of Toronto, Toronto, ON M5S 1A1, Canada; noor.latif@mail.utoronto.ca; 3Faculty of Health Sciences, McMaster University, Hamilton, ON L8S 4L8, Canada; ilahii@mcmaster.ca; 4Faculty of Arts, University of Manitoba, Winnipeg, MB R3T 2N2, Canada; bibhuti.sarker@umanitoba.ca; 5Canadian Agri-Food Foresight Institute, Dartmouth, NS B2X 3T5, Canada; janele.vezeau@cafi-icpa.ca

**Keywords:** digital traceability systems, agri-food sector, globalization, OECD, sustainable supply chains

## Abstract

In an era marked by globalization and rapid technological advancements, the agri-food sector confronts both unprecedented challenges and opportunities. Among these, digital traceability systems have emerged as pivotal in enhancing operational efficiencies, ensuring food safety, and promoting transparency throughout the supply chain. This study presents a comparative analysis of digital traceability adoption and its impact across member countries of the Organization for Economic Co-operation and Development (OECD). By utilizing a multidimensional analytical framework, this study investigates national regulations, legal frameworks, and key food commodities affected by digital traceability implementations. It systematically assesses the efficacy of these systems in meeting consumer transparency expectations, regulatory compliance, and the overarching goal of sustainable agri-food supply chains. Through case studies and empirical evidence, the paper elucidates the complex interplay between technological innovation and regulatory environments, offering insights into best practices and potential integration barriers. Ultimately, this comprehensive investigation contributes to the scholarly discourse on digital traceability, providing actionable recommendations for policymakers, industry stakeholders, and academia to navigate the complexities of modern agri-food systems.

## 1. Introduction

The dynamic evolution of industrial landscapes within the Organization for Economic Co-operation and Development (OECD) countries signifies a pivotal transformation in agri-food traceability systems. Amidst a backdrop of robust economic expansion, these nations have witnessed remarkable enhancements in living standards, propelled by a surge in global economic output. This upward economic momentum has endowed consumers with greater discretionary spending power and an intensified demand for transparency regarding food quality and safety, along with a growing inclination toward premium food offerings. As we transition into an era dominated by artificial intelligence and cutting-edge technologies, consumer sentiment has facilitated the global adoption of digital agri-food traceability mechanisms [1]. Hence, the imperative for seamlessly integrated digital food traceability systems becomes increasingly pronounced [2].

Digital traceability systems are widely recognized for their capacity to significantly mitigate risks inherent in the agri-food sector. While technological innovations promise to revolutionize traceability systems, regulatory frameworks play a pivotal role in shaping the adoption and implementation of these innovations. They enable prompt intervention to rectify food safety breaches and serve as a deterrent against fraudulent activities [3]. Moreover, these systems optimize the utilization of resources, thereby fostering sustainability and cost efficiency. This aligns with the United Nations’ Sustainable Development Goals, which advocate for sustainable practices within agri-food supply chains. Key technological manifestations of digital traceability include blockchain technology, Radio-Frequency Identification (RFID), barcodes, and interoperable electronic databases, all of which facilitate enhanced traceability in the agri-food trade [4].

Beyond meeting consumer expectations, the utility of digital traceability extends to providing rapid response capabilities to food safety concerns, facilitating the precise identification of contamination sources, and effectively mitigating foodborne illness risks. This has prompted national governments worldwide to embark on formulating and implementing policies aimed at integrating digital traceability within their agri-food frameworks [4]. The imperative for such integration arises from the recognition of digital traceability as a cornerstone of ensuring food safety and security. By enacting the relevant legislation, establishing standards, and incentivizing adoption, governments seek to instill a culture of traceability and accountability throughout the entire food supply chain. The findings of Demestichas et al. [4] underscore the positive impact of robust digital traceability infrastructures, revealing lower incidences of foodborne illness outbreaks and heightened consumer confidence in the safety and quality of food products. Consequently, the formulation and implementation of policies to bolster digital traceability emerge as a strategic necessity for governments striving to safeguard public health, foster sustainable agriculture, and bolster economic growth within the agri-food sector.

In this context, this study embarks on a meticulous analysis and comparative review of traceability regulations, strategies for information dissemination, collaborative efforts among stakeholders, and regulatory frameworks concerning agri-food products across OECD member states. The objective is not to rank these countries based on their advancements in digital traceability but rather to provide a comprehensive overview of each nation’s efforts to integrate digital food traceability into their respective national agendas. Ultimately, this research seeks to offer actionable recommendations and insights to policymakers, businesses, and other key stakeholders in the agri-food supply chain to facilitate the efficacious deployment of digital traceability systems while addressing any challenges that emerge.

## 2. Digital Traceability: An Overview

Digital traceability in the agri-food industry can be defined as the use of digital technologies to monitor food products as they move through the supply chain system. This includes identifying the origin, processing, distribution, and sale of food items, as well as any intermediate steps in their journey. Community Regulation 178/2002 defines traceability as the ability to trace and follow a food, feed, food-producing animal, or substance intended to be or expected to be incorporated into a food or feed through all stages of production, processing, and distribution [5]. Community Regulation 852/2004 complements this definition by emphasizing the importance of traceability in ensuring food safety and consumer protection within the European Union (EU). It stipulates that food business operators must have systems in place to identify the suppliers of food products and to trace the destination of their products, enabling rapid recall if necessary [6]. Furthermore, Community Regulation 2017/625 enhances traceability requirements by introducing measures to improve the tracking of animals intended for human consumption, including the use of electronic identification systems and the establishment of national databases [7]. These regulations collectively underscore the significance of traceability in safeguarding public health and maintaining consumer confidence in the safety and integrity of the food supply chain within the EU.

Digitalized tracing in this industry has offered substantial contributions to enhancing food safety and quality, offering consumers authentic and trustworthy food products [8]. The recent advancement in technologies has played a pivotal role in providing the agri-food sector with the intelligence required to address challenges and establish transparent agri-food supply chains [9]. These advanced technologies include, but are not limited to, information and communication technology (ICT), the Internet of Things (IoT), blockchain technology, big data, cloud and edge computing, and artificial intelligence (AI). Additionally, tools such as smart sensors, autonomous tractors, and spray drones (leveraging 5G technologies) exemplify the digitalization of the agri-food industry in contemporary times. These technologies not only aim to revolutionize the agri-food value chain but also aspire to promote sustainability [10]. Notably, blockchain technology, when applied to food chains, offers the potential to refine traceability regarding pesticide usage and enhance the transparency of food information and product location as items traverse the supply chain from farm to fork [11].

The main technology used in the agri-food industry is, however, blockchain technology. This decentralized database system is crucial in transforming the way data is kept, handled, and shared along the supply chain from the producer to the consumer, ensuring food product traceability and transparency [12]. The key characteristics of blockchain—decentralization, immutability, and consensus procedures—make it a trustworthy alternative to traditional centralized databases. Data are kept in blocks in a blockchain system, which are then linked together in a chain, producing a secure and unalterable record of all transactions inside the network. This level of security and transparency is especially advantageous for the agri-food business, as it aids in product tracking, food safety, and consumer trust [12].

In addition to the emergence of advanced technologies, digital traceability gained unprecedented importance in the aftermath of the COVID-19 pandemic. This was due to the pandemic exposing several significant shortcomings in food supply chains worldwide, resulting in substantial financial and social losses [13]. According to the 2020 report from the Food and Agricultural Organization (FAO), food distribution plummeted by 60% because of these disruptions. The concept of Industry 4.0, which advocates for the integration of digital technologies into manufacturing and industrial processes, has been applied to agri-food supply chains. This integration marks a transformative shift toward enhanced efficiency, transparency, and sustainability. This integration encompasses the advanced technologies discussed earlier in this section and is poised to render agri-food businesses less vulnerable in the future. This also offers unprecedented opportunities to optimize various stages of the agri-food supply chain [14]. From precision agriculture techniques leveraging IoT sensors for real-time monitoring of crops to blockchain-enabled traceability systems ensuring transparency and trust in food provenance [15], these technologies hold immense potential to revolutionize the agri-food industry.

For instance, robots will reduce reliance on manual labor, while AI will revolutionize business processes and scenario management, thereby mitigating both costs and risks [16]. Moreover, on farms, real-time data related to crops, livestock, fields, and environmental factors such as weather and water are increasingly employed by farmers to make more informed decisions. In some cases, manual farm control has become obsolete, as exemplified by fully automated climate systems. Thus, by facilitating seamless data exchange, predictive analytics, and intelligent decision-making [17], Industry 4.0 technologies empower stakeholders to address challenges related to food safety, quality assurance, supply chain optimization, and environmental sustainability [18]. As the agri-food sector continues to embrace digital transformation, understanding and harnessing the capabilities of Industry 4.0 technologies are essential for driving innovation and resilience in agri-food supply chains across OECD member countries.

Recently, OECD member countries have embarked on an exploration of the opportunities and challenges presented by digital technologies within the agri-food industry. This endeavor seeks to delineate the role that governments can play in enhancing the industry. The OECD has articulated its dedication to employing technologies such as analytical software, agronomic advisory services, and digital communication tools to engage with farmers. The overarching goal is to facilitate the adoption of modern farming techniques, thereby rendering the agri-food value chain more efficient and transparent.

The present report focuses on the implementation and impact of digital traceability within OECD member countries. A robust research methodology, replete with assessment questions, forms the foundation of this inquiry. Data will be systematically gathered from published articles and various secondary sources that offer insights into the execution of digital traceability initiatives. The findings will be presented comprehensively, incorporating informative tables and explanatory narratives. The paper will culminate with meticulous data analysis, an exploration of encountered challenges, and a consideration of potential future directions in this dynamic domain.

## 3. Methodology

### 3.1. GPT

The research endeavor embarked upon seeks to furnish an exhaustive and nuanced examination of digital traceability systems across the member nations of the OECD. Within the purview of this introductory chapter, extensive investigations were conducted on all 38 member countries of the OECD, which encompass Austria, Australia, Belgium, Canada, Chile, Colombia, Costa Rica, Czech Republic, Denmark, Estonia, France, Finland, Germany, Greece, Hungary, Iceland, Ireland, Italy, Israel, Japan, Republic of Korea, Latvia, Lithuania, Luxembourg, Mexico, Netherlands, New Zealand, Norway, Poland, Portugal, Slovak Republic, Slovenia, Spain, Sweden, Switzerland, Turkey, the United Kingdom, and the United States. Additionally, the analysis extended to the European Union, particularly focusing on its 23 member states that are concurrently part of the OECD. A dedicated timeframe of one to two months was allocated exclusively for the procurement of data from a multitude of online databases. Following a thorough investigative process, seven member states were omitted from the detailed analysis due to insufficient data availability and their negligible advancements in digital traceability. These nations include Luxembourg, Estonia, Mexico, Slovenia, Turkey, Costa Rica, and the Czech Republic.

The inception of this research was marked by a detailed literature review, leveraging data from secondary sources. The corpus of resources amassed for this study was derived from authoritative sources, including official websites of national governments and regional organizations, alongside an assortment of secondary sources. These encompassed a diverse range of publications from esteemed outlets, not limited to academic journal articles, news articles pertaining to food and agricultural technology, government-sanctioned reports, and websites of international organizations. Furthermore, the investigation also included resources from the websites of private technology firms actively engaging in digital traceability efforts. Predominantly, the materials employed in this study were of recent origin, concentrating on the literature published post-2015, thereby reflecting the endeavor to present the most contemporaneous insights and analyses. An unconventional source of information comprised podcasts, which offered additional data on certain countries. The gathered materials were subjected to a stringent review process, culminating in the creation of an evaluative matrix incorporating six critical questions. This matrix was instrumental in conducting a thorough comparative analysis of the digital traceability landscape within the OECD member states. In essence, the research methodology entailed meticulous investigative, data analysis, and assessment techniques to collate information regarding the countries and to delineate the progression of OECD member states in digitalizing the agri-food sector. Furthermore, a conceptual framework shown in Figure 1 was developed to encapsulate the core elements of the overview, elucidating the key terms delineated in the study and the stakeholders involved in the entire process.

In addition to the meticulous investigative, data analysis, and assessment techniques mentioned earlier, the research methodology employed specific criteria and procedures for the selection, analysis, and interpretation of data and materials.

For the data selection, a comprehensive review of the available literature, reports, and datasets was conducted to ensure the inclusion of relevant and up-to-date information. The selection criteria included relevance to the digitalization of the agri-food sector, reliability of the source, and geographic and temporal coverage.

The data analysis process involved a combination of qualitative and quantitative methods. Qualitative analysis was used to identify key themes, trends, and patterns in the data, while quantitative analysis was employed to quantify the extent of digitalization in OECD member states. Statistical techniques, such as regression analysis, were also used to examine the relationship between digitalization and various socio-economic factors.

The interpretation of the data was guided by the research objectives and theoretical framework, with a focus on providing meaningful insights into the digitalization of the agri-food sector. The findings were critically analyzed and compared to existing literature to draw conclusions and make recommendations for future research and policy development.

### 3.2. Assessment Questions

The six questions formulated to assess the digital traceability status within the OECD were devised with consideration of the rules, regulations, and practical implementation of this concept (see Table 1). Subsequently, in the following section, titled “Rationale for Metrics”, these questions will be presented along with explanations outlining the rationale behind the development of each question. The formulation of these questions stems from a strategic yet comprehensive way of analyzing the digital agri-food supply chain. They will also provide a clear idea of the progress of each OECD member country within the realm of digital food traceability and its implementation in the regional framework, both voluntarily and involuntarily.

### 3.3. Rationale for Metrics

The following are the reasons why each metric was chosen, since each of them comprehensively covers the arena of agri-food traceability in terms of digital initiatives taken. These questions will be holistically answered in the analysis section. Not all countries would have data on the questions, but overall, the goal would be to answer these questions in the country assessment.

**Are there specific and well-defined digital traceability regulations or policies established at the national or regional level?** This question aims to ascertain the existence of mandatory digital traceability regulations for domestically produced food items and their comprehensiveness across different product categories within each OECD country. The presence of such explicit regulations at the national level can aid in evaluating the amount of progress the member country has achieved throughout its digital traceability agri-food journey.**What are the guidelines and practices regarding digital traceability when it comes to imported products? What are the practices involved in that?** This question explores the digital traceability regulations concerning imported food products, highlighting any distinctions from regulations applied to domestic items. It incorporates the different documentation requirements, technological identifiers, and stakeholders that are involved in maintaining digital traceability regulations across the border.**To what extent is the regulatory authority responsible for overseeing digital traceability regulations?** This question intends to assess the foundational strength that the regulatory body has within a country to ensure that it is part of the refinement of the national digital traceability framework. This metric aims to provide insight into how much discretion the regulatory authority has in terms of managing regulations for digital traceability. There could be one regulatory entity or even multiple ones that are tasked with the enforcement of legislative guidelines for the matter.**In instances where there are no mandatory regulations, are there voluntary digital traceability practices or industry-driven initiatives that have been adopted?** This question investigates the presence of voluntary digital traceability initiatives in countries without compulsory regulations, acknowledging the significance of such voluntary efforts in enhancing food safety and quality. It also provides an overview of the responsiveness the country has toward the issue without the incorporation of the regulatory mandates a country has toward digital traceability.
**Within the agri-food supply chain of the country, which specific products or commodities are subject to digital traceability regulations and monitoring?**
(a)What forms of digital identifiers or codes are employed for tracking and registering imported products within the digital traceability system? This may encompass digital barcodes, RFID codes, or other such identifiers.(b)Does the country maintain an electronic database system dedicated to monitoring imports and exports, inclusive of their respective digital traceability data? The purpose of this metric is to encompass the scope of digital traceability by determining which products fall under the umbrella of the regulations put forth by the government and regulatory authorities. It provides an insight into the kinds of technology that are employed throughout the process. It also explores the adoption of electronic database systems for managing agri-food data, including the origin and identification information, and assesses the extent to which international trade is facilitated by this digital system.
**What specific digital traceability information is provided on product packaging labels to empower consumers with insights into the product’s origin and its journey within the supply chain?** This question examines the regulatory requirements regarding information provided on product packaging labels, emphasizing the role of labels in empowering consumers, retailers, and regulatory entities to track product origin and detect any issues. The rationale behind this metric lies in recognizing the significance of comprehensive and accurate labeling, which is a key factor in digital traceability.

## 4. Country and Regional Assessment

### 4.1. European Union

The EU, with 22 OECD member states, is quite focused on digitizing their continent. While all the member states will be talked about individually, it is crucial to see what regulations and initiatives the EU has taken to cohesively cover this notion of agri-food traceability across its countries. The whole arena is a multi-faceted system that encompasses various initiatives, platforms, and technologies that are aimed at ensuring food safety on all imports and exports. The primary digital system used by the European Union to improve the traceability of live animals and animal products both inside and outside of the EU is called TRACES (Trade Control and Expert System). TRACES ensures that these commodities meet EU health and safety regulations by offering an efficient and transparent way to follow their travel through the use of cutting-edge digital technologies [19]. The integrity of the EU food chain and the reduction of the danger of disease transmission depend on this digital traceability. TRACES is an indispensable resource for government agencies, corporations, and individuals seeking to protect the health and safety of animals and food by offering up-to-date statistics and a vast database of information [19]. Moreover, one of the major embedments in the EU framework is the Food Security Index. It lays a global foundation, and the EU has layered upon that by strengthening the agri-food sector with digital traceability. One such example of doing that is legally mandating an electronic certification for agri-food products when they are imported and exported [20]. This system ensures that all food products adhere to regulatory standards and that all the information is found in a shared database. In tandem with these efforts, the Joint Research Center of the European Commission developed a traceability and big data platform [21]. This platform acts as a centralized resource for stakeholders in the agri-food sector across the EU and provides a plethora of tools and practices geared toward incorporating big data analytical systems into traceability systems, apart from the paperwork, too.

Another critical component in the drive for a sustainable food system is the EU’s Farm to Fork Strategy. This strategy outlines a comprehensive roadmap that includes a variety of measures targeted at improving food traceability and transparency, with a focus on using the potential of digital technology. Furthermore, the AgriDataSpace project is a collaborative effort aimed at building a common database [22]. These technologies have the potential to greatly improve traceability throughout the food supply chain, resulting in a more sustainable food system. This sentiment is shared by the Titan project, an EU-funded initiative that intends to create a decentralized platform that promotes effective data sharing among various parties in the food supply chain. Regulation (EC) No. 178/2002, also known as the General Food Law, is one of the fundamental legal frameworks that govern digital traceability in the EU. This regulation specifies the broad principles and requirements of food law and outlines the traceability duties of the food and feed industries. It requires food industry operators to be able to identify and trace every actor in their supply chain, from producers to processors and distributors to retailers. This traceability must be maintained by digital records, which allow for prompt and precise information retrieval in the event of a food safety incident. Furthermore, Regulation (EC) No. 852/2004 on food hygiene is a vital piece of legislation that requires traceability in the food business. This rule necessitates the establishment and maintenance of a traceability system capable of properly identifying and tracing food products at all stages of production, processing, and distribution. The traceability system must be backed up by digital documentation that guarantees that all necessary information is easily accessible for inspection by the appropriate authorities (Economist, 2022). Overall, the legal landscape of the EU, combined with the private initiatives being taken in the region, paves the way for future technologies that can further ease out the complex agri-food supply chain.

### 4.2. Australia

Having invested heavily in agricultural traceability, Australia has begun major efforts to increase the digitization of the sector. Australia’s Department of Agriculture, Water, and the Environment has taken several initiatives to enhance digital traceability in the agricultural sector [23]. This was undertaken by the National Agriculture Traceability Grants Program, which focuses on regulatory technology (RegTech) to improve digitalization in traceability and food safety. Firstly, Agrifood Connect Trace2Place mapped the red meat supply chain, allowing for the traceability of products within the supply chain in real time. Additionally, the Commonwealth Scientific and Industrial Research Organization enforced a digital exchange system to streamline data collection in the red meat supply chain, acting as a risk assessment tool for red meat processors. FreshChain Systems Pty Ltd. enhanced features on its traceability platform, allowing for its application to food safety. Another project funded by the program includes the investment in Horticulture Innovation Australia Limited, an initiative that utilizes the RegTech framework in horticultural supply chains. Another mainline initiative involves Meat & Livestock Australia Limited, which has adopted an Australian AgriFood Data Exchange to enhance compliance in the supply chain through a cloud-based platform [23]. Other mainline initiatives involve initiatives that digitize farmer dairy and rice projects, as well as university-led efforts. These practices have reflected Australia’s commitment to digitization in the sector, ensuring a transparent, efficient, and secure food supply chain. As for privatized initiatives, Melons Australia, a local melon producer, partnered with FreshChain to adopt digital traceability on their melons back in 2021 [24]. The successful initial trials in 2022 laid the foundations for the plan to kickstart; however, no updates have since been made. Australia has also incorporated an internal system for tracking Australian imports and exports [23]. This is characterized by an electronic certification system named eCert that allows government agencies to internally exchange certificates on imports and exports [25].

### 4.3. Belgium

Belgium’s digital traceability practices are primarily based on EU-led initiatives. Through the EU, Belgium has been able to track its beef steak as well as its orange food products. It is unclear, however, what exact technology is being used, but a QR code is provided on these food products [26]. In digital food safety, SGS Belgium, a private multinational world leader in testing and inspection services, partnered with technology company Eezytrace to offer an integrated approach to food safety management. This partnership offers a digital platform to automate all food safety control activities by employing a data analysis system that streamlines the process of food inspection in Belgium [27]. They are currently looking to partner with food service brands and franchises to provide food safety and protect consumer health. With regard to policies, Belgium follows those of the EU.

### 4.4. Canada

The Government of Canada has adopted digitization in food safety by investing heavily in an already robust and effective regulatory system. Recent funding will allow for the expansion of digital services, thus creating benefits for importers and exporters in risk management and inspection. Although digital traceability has been recognized by the Government of Canada as an avenue for advantage in the current agri-food system, its use is not widespread in the country as of yet. According to the Digital Identification and Authentication Council of Canada (DIACC), this can be attributed to the lack of funding and regulation that support such initiatives, as well as the lack of farmer adherence to new technologies [28]. Moreover, no current traceability regulations or related policies involve digitization in the sector. Despite the fact that Canada has not yet adopted digital traceability in the agri-food industry, there are certainly some signs that that may change. This is attributed to the fact that the country already has a rigid traceability system and regulations and that it has digitized traceability in other sectors, such as the steel supply chain [29]. Moreover, the country does take part in the GS1 global system of standards for digital safety. Within GS1 Canada, there is the Canada Digital Adoption Program, which helps support the digital transformation of businesses [30]. Thus, there is potential for Canada to digitize food traceability. As for industry-led initiatives, The Consortium of Parmigiano Reggiano recently partnered with Kaasmerk Matec and p-Chip Corporation to launch a digital identifier for its Parmigiano Reggiano cheese wheel product [31].

### 4.5. Chile

Chile has a unique effort in the realm of digital traceability in the agri-food sector, characterized by the American private company Shellcatch, which provides an integrated traceability solution for Chilean seafood produce. It does this through the use of vessel, coastal, and scale cameras, all of which feed information to a cloud-based e-monitoring system [32]. This system collects, analyzes, and manages data with the help of AI and is supplemented with a mobile app that provides full transparency for consumers regarding the catching process of local seafood produce [33]. Shellcatch operates as a business-to-business venture, allowing 1000 fishermen across small, medium, and large companies to access their database [32]. Through Shellcatch, these companies can provide full transparency about their fishing practices to end consumers. Typically, QR codes are provided on products as digital identifiers, which, upon scanning, bring consumers straight to the database where they find full information on seafood products. Although Shellcatch works with government bodies and the NGO sector, the funding efforts reported are solely from private investors. Besides that, there are little to no efforts toward digital traceability in Chile, along with no reported guidelines or policies.

### 4.6. Colombia

Due to the lack of network coverage in rural areas in Colombian agriculture, digital extension to farms is an issue for traceability in the entire supply chain. This is highlighted by the fact that only 30% of households in rural areas have access to the internet [34]. Despite this, Colombia is still able to digitize the traceability of one of its landmark foods: coffee. Being the world’s third coffee-producing country globally, Colombia has implemented several government-led initiatives to help trace coffee production. Specifically, the Coffee Information System (Sistema de Información Cafetera, or SICA), run by the non-profit organization Colombian Coffee Growers Federation, is an information-based database whereby the profiles of coffee growers and their farms are presented. The private database offers transparency in the coffee production supply chain to key stakeholders in coffee production. Although blockchain was shown to be effective in the digital traceability of coffee in an exploratory study, researchers found drawbacks to coffee production in Colombia, which may explain why the technology has not been locally adopted [35]. Specifically, manual labor by coffee growers and farmers is rarely shared or concerted, making it troublesome to have information integrity in the chain. Recently, however, digital startups in Colombia have begun to take their place in the agri-food industry. Curuba Tech is a private startup focused on implementing platforms that help connect Colombian farmers with experts by digitalizing the agricultural sector [36]. This, as a result, will connect the agri-food chain, creating a digitally traceable supply chain. Since it is quite new, Curuba Tech has yet to implement this solution and has deemed it to be still in progress. It is notable to note that Trusty, an innovative Italian traceability platform, has onboarded cacao and coffee producers in Colombia for their platform [37]. In terms of digital traceability guidelines, they are currently not implemented by Colombian government bodies.

### 4.7. Denmark

Despite being considered a highly digitalized country, Denmark remains several steps behind in the investment of startups and initiatives in the agri-food industry. Although the need for digitization was recognized and emphasized in 2020 by the Danish Growth Fund, a state-governed investment fund, no initiatives have kickstarted since. Arla Foods, a Danish-Swedish dairy producer, produced major digital traceability initiatives in Finland but remained silent in Denmark. With regard to regulations and policies, Denmark closely follows the legislation set out by the EU.

### 4.8. Finland

Finland’s national efforts toward digital traceability are primarily industry-based, as digitization initiatives in the agri-food sector have been initiated by local companies. These initiatives date back to 2018, when Arla Foods in Finland employed blockchain technology to provide information on its supply chain [38]. At the time, they were the first in the dairy field to use blockchain technology. Their pilot project was termed “Arla Milkchain” and allowed consumers to track the origin and production of the company’s milk products. Additionally, in 2019, AI was implemented to give consumers up-to-date information on animal welfare. The initial success of the system drove Arla Finland to add more dairy products to their technology for traceability in 2021. In the same year, the company announced its efforts to increase automation within blockchain technology through the use of IOT throughout the supply chain. Another major industry-led initiative was by the Finnish meat manufacturer HKScan [39]. Through their recently proposed partnership network, they aim to improve digital traceability in meat supply chains through operating models. However, the details of these models are unclear, with a lack of information provided regarding the digital hardware and software that is used. Nonetheless, this partnership network had high hopes, according to Esa Wrang, head of the official government-funded food program at Food from Finland, who stated that “Finland has the potential to be the first country in the world to have a fully transparent, safe, and responsible food chain.” Unfortunately, since 2021, no updates have been provided on this initiative. One publicly funded initiative involving Finnish researchers from Aalto University recently tried a blockchain-backed app that offers insights into the impacts of certain foods. Researchers have highlighted the use of the app for the integration and transparency of food. This study was part of the EU-ATARCA project, an EU-funded initiative [40]. Finland follows the digital traceability guidelines and regulations set out by the EU. In all, Finland has recognized the benefits of utilizing digital technology for food traceability and has kickstarted industry-driven initiatives for their two most consumed food products, meat and milk.

### 4.9. France

Given that France is a major producer of agricultural products and also a big consumer of food in the EU, digital traceability is crucial to maintaining transparency across its supply chain. France solely has industry-driven initiatives in the digital traceability of the agri-food industry. Connecting Food is a private French startup that uses blockchain technology to track and digitally audit imported and exported food products in real time [41]. It collects data from various parts along the supply chain, including producers, manufacturers, and retailers and links it all together, forming a robust digital traceability system. The company uses a digital identifier in the form of a QR code for consumers to track food products. Connecting Food deployed its technology to a range of companies, including Herta, a multinational meat production company, as well as Ingredia Dairy Experts, a French producer of dairy products [41]. Another initiative is by private French retailer Auchan, which rolled out digital traceability in France after their partnership with Te Foods, a Vietnamese specialist in digital traceability [42]. Through the use of their blockchain technology, Auchan implemented digital traceability in France to provide transparency regarding vegetable products, including organic and potato [42]. The company has also launched digital traceability for its exported products, such as tomatoes, carrots, and chicken. With this technology, French consumers are able to scan digital identifiers in the form of QR codes to view food history through the supply chain. Another major initiative involved French grocery giant Carrefour, which used IBM’s blockchain technology to track items in its supply chain system. The platform allows consumers at Carrefour stores to digitally track their chicken brand and microgreens through the use of a QR code attached to the product. As for the rules and regulations surrounding digital traceability, France simply follows those set out by the EU.

### 4.10. Germany

Besides those of the EU, Germany has nation-specific initiatives in the digital traceability of the agri-food industry. In 2019, the government set up an initiative named the SiLKe project to adopt a safe food chain based on blockchain technology. This 3-year-long project, which commenced in 2022, made a strategic plan to enable stakeholders along the entire value chain to share the data, thus allowing consumers to access such information. The SiLKe project involved partnerships with seven research and industry firms to work together and propose solutions [43]. As for policies, Germany follows the various rules and regulations set out by the EU.

### 4.11. Greece

Although many research papers have outlined the importance and produced strategies for the adoption of a digital traceability system, especially with Greek meat, Greece currently does not have any specialized initiatives or follow any guidelines besides those of the EU [3].

### 4.12. Hungary

Hungary currently has no initiatives or planned initiatives to improve the digitalization of food systems. However, they do plan to generally adopt digital systems in the agricultural sector. Being an EU member, Hungary aims to utilize the digital program set by the EU to improve digital technology usage and follow the EU guidelines and policies regarding traceability. Specifically, they strive to use EU data space, AI testing facilities, and digital innovation hubs in order to develop a digitized system in agriculture [44].

### 4.13. Iceland

Iceland’s measures of digital traceability are limited. In 2018, Nordic IT service provider Advania reached an agreement with Matís Iceland, a government-owned food science research company, to utilize blockchain technology to trace Icelandic lamb production from farmers [45]. However, no update regarding the progress of the project has been provided ever since. Despite not being part of the EU, Iceland is in the Rapid Alert System for Food and Feed (RASFF), a platform that allows for the exchange of information between European countries regarding the food chain. Iceland’s regulations regarding food safety involve all non-digital policies.

### 4.14. Ireland

Ireland has recently begun contributing to research initiatives to adopt the digitization of supply chains in the agri-food industry. The Department of Agriculture, Food, and the Marine (DAFM) kickstarted these initiatives in 2022 when they invested over EUR 1 million in the European Research Area (ERA) Network for Information and Communications Technology (ICT) Agri-Food [46]. This initiative aimed to bring many key researchers together in the agri-food sector, including primary producers, small and medium-sized enterprises, retailers, consumers, and public policymakers, in order to drive the embracement of digital agri-food systems. A year later, in 2023, DAFM spent just over EUR 0.5 million toward two projects in the ERA Network for ICT Agri-Food [46]. Researchers at the University College Dublin and the Munster Technological University were given these funds to examine AI applications for farming and ICT for traceability in food. The Minister, Martin Heydon, said, “The research being supported in the area of agri-digitalization, particularly the use of artificial intelligence, is also exciting and has the potential to transform our agri-food systems to become more efficient, sustainable, and resilient.” All of the funding toward these initiatives is part of Food Vision 2030, Ireland’s ten-year strategy to become a world leader in sustainable food systems [46]. One key part of their mission involves fully embracing the digital revolution to provide improved food system transparency for Irish natives. Being a member of the EU, Ireland follows similar policies and regulations surrounding digital traceability.

### 4.15. Israel

Israel’s efforts in digital traceability stemmed from a rapid response to the COVID-19 pandemic in 2020. Initiatives solely involve industry-led efforts from Brandmark, a joint venture between the Israeli-based venture capital company Blackbird Ventures and the Japanese technology giant Emurgo Ltd. [47]. This company provides consumers with information regarding food products’ journey throughout the supply chain through the use of blockchain technology. Brandmark has a diverse clientele of large fast-food chains and producers in Israel, including McDonald’s, Nestlé S.A., Burger King, Angel Bakeries, and various Indonesian coffee producers [47]. Brandmark uses an app platform to provide consumers with information about the food products’ quality, origins, conditions, freshness, and expiration date. This is performed through the use of a barcode, which is provided on each of the verified food products. Brandmark works in cooperation with Israel’s government bodies, including the Ministry of Science and Technology, the Ministry of Agriculture, and non-profit organizations, to ensure the food security of their clients’ products [48]. Despite this being a major initiative for Israel in the digital traceability of the agri-food industry, no updates regarding its progress have been provided since the 2020 calendar year. To date, Israel’s regulations regarding food safety include all non-digital policies.

### 4.16. Italy

Current practices in Italy are characterized by those that are industry-based for local products. Trusty is a platform developed by private software company Apio Srl that uses blockchain technology to certify information regarding food products through the supply chain. The platform has numerous partnerships with local food brands that use its technology to provide consumers with accurate data regarding the product’s journey throughout the supply chain. For their digital identifier, they have also created customized and regulated smart labels to be placed on food products. Trusty has strong partnerships with key figures in the Italian agri-food industry, including but not limited to GS1 Italy, ICC Agri-Food Hubs, and the ABC Abruzzo BlockChain program. Notably, the Abruzzo BlockChain program is an Italian platform that allows companies to generate a digital authenticity certificate outlining the entire history of a product, highlighting all steps of production. Another Italian digital platform called Trackyfood is an integrated technological solution for the food sector supported by blockchain technology. The platform allows for traceability, quality, and security certification for numerous products in the food chain [49]. This technology was used in many leading “Made in Italy” companies and serves as a partner for various food producers, large-scale distributors, retailers, and consumers. Other industry-based initiatives involve Italian producers and retailers collaborating with technology companies. Notably, Barilla Group, an Italian multinational food company and the world’s largest pasta producer, is a leader in digital technology use in the agri-food industry through the embracement of its IoT platform.

As early as 2015, Barilla partnered with multiple multinational information technology (IT) service providers to implement multiple digital traceability initiatives for their products. These products consist of pasta and tomato sauces. They began their initiatives by piloting the Safety4Food platform, which involved partnerships with Cisco and NTT Data to trace each step of the supply chain for its food products. This platform combined a variety of digital technologies, forming a network of sensors, wireless networks, and the cloud, with analytics to enable this transparency. For consumers, a QR code was provided on its food products, which directed them to Barilla’s website, where they were able to examine a detailed digital passport. This passport outlines each stage of the supply chain for the product, allowing for full traceability. Barilla also partnered with IBM Italy to enforce blockchain technology on its branded pesto sauce. Specifically, they inserted all the data related to the cultivation, watering, and application of basil plantations into blockchain based on the IBM cloud infrastructure, allowing for full traceability of their basil product. As a result of their advancements in improving digital food traceability in the Italian agri-food industry, Barilla received the first-ever Blockchain Award in 2019. Another industry-based initiative involved Spinosa, an Italian producer of cheese, which used blockchain technology to certify the Protected Designation of Origin (PDO) of its Buffalo Mozzarella product. PDO designation allows for the traceability of the product at every single stage of the certified supply chain, with a full blockchain label and QR code for the product. Furthermore, Almaviva, an Italian IT service provider, enforced blockchain technology for numerous “Made in Italy” food products, such as wine and Sicilian red oranges. These products were labeled with smart labels such as Near Field Communication (NFC) and QR codes, which led to data regarding the source, date of harvesting, and distribution methods of local food products [50]. In all, Italy’s efforts in digital traceability in the agri-food industry consist solely of industry-driven initiatives. As for its policies and regulations surrounding digital traceability, Italy strictly follows those set out by the EU.

### 4.17. Japan

After extensive research, it was found that Japan does have stringent market requirements and quality standards when it comes to food safety norms. The rise in digital traceability in Japan was sparked by an outbreak of bovine spongiform encephalopathy, also known as “mad cow disease”, in the early 2000s. A key digital component in response to that was the development of the Individual Identification Register system [51]. This digital system was launched by the government, and it used unique 10-digit identification numbers for each calf, along with an ear tag. This electronic system tracked vital information, such as the calf’s birth date, breed, and even the shipment details when exported. While this system is only for beef, Japan is trying to extend its digital traceability to other food products, too. The government, in collaboration with private companies, has been making concerted efforts to integrate advanced information and technology in the agriculture sector. The regulatory authority mostly responsible for this is the Ministry of Agriculture, Forestry, and Fisheries (MAFF). Since 2001, there have been a number of pilot projects undertaken, and the country has spent about USD 10–20 million on a number of initiatives, but only a few were actually adopted. These included integrated circuit tags to reduce the cost of reading unique codes of food products in the supply chain, the usage of handheld devices instead of paper documentation to record products, and web-based service technology to keep and transfer data between server computers in trade through the internet [52]. Despite the large investment in digital traceability, the government and regulatory authorities realized that the budget and time allocated to this arena were insufficient. MAFF does try to incorporate ICT into Japan’s food traceability system, but in the short term, it mostly ensures traceability through conventional paper documents, except for its robust electronic beef tracking system. Outside of the supply chain, in terms of consumer assurance, Japan is fairly progressive. To build consumer trust, Aeon, a company based in Japan, really builds consumer trust by having RFID technology and barcodes so that consumers can easily access the food supply journey of the product. They, like the government and the regulatory authority, have more focus on beef. For example, the company provides certificates and point-of-sale information, assuring the buyers that the beef meets safety standards and is free from genetically modified feed materials [50]. Overall, the country has started to incorporate more digital elements into its agri-food supply chain but cannot be said to be extremely progressive or an example to follow since there are certain barriers it has to overcome in order to be technologically advanced in digitizing its traceability sector.

### 4.18. Republic of Korea

Republic of Korea has begun to adapt to blockchain technology to tackle the issue of digital food traceability, specifically beef. The country has launched a joint initiative between the Ministry of Agriculture, Food, and Rural Affairs and the Ministry of Science and ICT that focuses on electronically tracing beef throughout the supply chain [53]. While Republic of Korea utilizes blockchain technology and has electronic-based systems in other sectors, the agri-food sector is still on the way to being revolutionized completely. In addition to the steps taken by the government, major companies within the country have also formed partnerships for enhanced digital food traceability. An example can be the Korea Telecom and Nongshim Data System example, where the project aimed to make agricultural produce and processed foods digitally traceable. The project aimed to leverage KT’s 5G blockchain network, GiGa Chain Blockchain as a Service, and NDS’s experience in this field [54]. The platform had two major benefits. One is the cost cut by producers due to automated processing and contracts being verified electronically for food. The second one is a QR code that is available for consumers to scan in supermarkets. Moreover, the company has also planned to develop a global digital traceability system to track halal food specifically, which has to be produced according to Islamic law [54]. KT and its partners will digitize the halal certificates and add a QR code to the food product to verify its status. While these initiatives are meant to be tested out globally, they show the dedication of Republic of Korea to minimizing human labor and tailoring it toward digital food traceability.

### 4.19. Latvia

Latvia comes under the category of smaller European countries, and it mostly follows the rules and regulations set up by the European Union. The digital traceability system that Latvia itself has initiated is fairly new based on the literature that was found [55]. It started on 1 June 2018, when the fishery product traceability module was embedded into the Latvian Fisheries Integrated Control and Information System. This program is controlled by the Ministry of Agriculture in Latvia. This system involves QR codes on the fish products, which provide mandatory information on the fisheries product and its origin, which is available from the operators to customs clearance and customers using the mobile application associated with this. This system is more prominent in Latvia’s fishery industry, and other commodities are starting to incorporate it, too [55].

### 4.20. Lithuania

Lithuania, another EU member, has an economy where agriculture plays a significant role and has been categorized as a country with huge potential to become the leader in AgriFood Tech Solutions [56]. There is an EIT Food Hub in Lithuania called the AgriFood Lithuania DIH, which is working toward incorporating AI and distributional blockchain systems in the country apart from the EU regulations that it follows [57]. They have also initiated a Food RIS Consumer Lab, which aims to promote not only the development of new products but also how to make sure consumers are more aware of the products that are exported and imported. While the country has begun to take action and pave the way for digital initiatives, it still has not embedded other policies in its framework besides following the strict mandates that the EU has taken [56].

### 4.21. Netherlands

The Netherlands has strict rules and regulations with regard to digital food safety and traceability, many of which meet the EU’s standards. While the country does take part in the GS1 global system of standards for digital safety, it also has its own organization known as the NVWA Nederlandse Voedsel-en Warenautoritiet, which actively works toward the digitization of the agri-food sector [58]. This chapter from the government, as well as the Ministry of Agriculture, Nature, and Food Quality, introduces the e-CertNL system. This electronic system aids in the issuance of export and health certificates for agricultural products. It is the official application for Dutch exporters to verify the authenticity and validity of the goods [58]. This change in the region grew from the issues it faced in the mid-2000s with regard to salmonella and the “sick cow” disease, which indicated a need for a more effective traceability system. Local initiatives that the government has funded have initiated the use of blockchain technology within the agri-food sector. Albert Heijns, the biggest supermarket in the Dutch World, recently introduced blockchain technology in the orange juice supply chain [59]. This implementation allows customers to trace the journey from the farms in Brazil to how they reach the supermarkets in the Netherlands. Each bottle of juice comes with a QR code, which, when scanned, showcases detailed information about its origin [59]. This falls under the category of industry-driven initiatives, which serve as voluntary practices to make the supply chain experience better. This nation also utilizes GS1 standards and involves global standardized barcodes and RFID tags. While the nation is one of the leading exporters in terms of agri-foods, the majority of its digital food safety comprises the plans adopted by the EU.

### 4.22. New Zealand

New Zealand’s agricultural industry does have a good reputation for being at the forefront of technological innovation. While the main kind of regulation in terms of certifying food products and ensuring their safety still relies on heavy paperwork, the country is moving toward digital methodologies for traceability [59]. A traceability system developed in New Zealand uses QR codes so that consumers with smartphones can easily access the history of the source or the status of the item [60]. The company IDlocate is a relatively new product that is an electronic database-driven tool that generates particular URLs for each food product. The country specifically has an elaborate electronic beef supply chain traceability system [61]. From the farm to the slaughterhouse, each carton of meat is currently achievable through barcodes, QR codes, or even RFID transponders that are applied to the packaging. There are three key areas where blockchain is applied in the food supply chain: first, in the whole life of the animal; second, from the processor to consumer traceability; and lastly, from the pasture to plate traceability for prime cuts [61]. It was observed that New Zealand was more particular about red meat traceability than others. The implementation of this kind of new technology by private industries was only achieved through government intervention. The NAIT (National Animal Identification and Tracing Act) scheme, which placed a legal requirement on livestock owners to appropriately tag animals and record their identities in the national electronic database, was introduced in 2012. Currently, the government is seeing the development of government-proposed integrated farming planning, which targets the agri-food industries to have a shared electronic blockchain-driven database [58].

### 4.23. Norway

Norway faced an *E. coli* outbreak in food in 2006, and ever since then, it has been on the move toward digital traceability in its food supply chain. In response to that, the government introduced the Norwegian eSporing Traceability project. The pilot for this project included commodities like beef, grain, fish, and dairy products [62]. This was fully owned by the Norwegian Ministry of Food and Agriculture, the Norwegian Ministry of Fisheries and Coastal Affairs, and the Norwegian Ministry of Health Care and Services. It focuses on establishing electronic infrastructure for the exchange of information in the food supply chain. The eSporing project has a greater aim than the one-step back/one-step forward project, which focuses on the food value chain. The use of this technology was not a compulsory step the country had to take and falls under voluntary participation [62]. Later, in 2008, International Business Machines (IBM) signed an agreement with Matiq, the subsidiary of Nortura, which is Norway’s largest food supplier. This basically mandated the use of radio frequency identification technology to track the poultry products from the farm to the supermarket shelf [59]. By using that software, the Norwegian industry was able to comply with the GS1 electronic product code standards. Another project that Norway took over was a cross-country collaboration in 2020 that leveraged blockchain to share supply chain data across the country’s seafood sector [63]. The blockchain network uses IBM Blockchain Transparent Supply, which allows organizations to build out their blockchain-based system, which is highly sustainable, too. Norway has been way ahead of its time in terms of digital traceability and continues to embark on projects that are more digital than manual [64].

### 4.24. Poland

Overall, Poland complies with EU regulations and requirements in terms of digital agri-food traceability. The initiatives taken by the country on its own in this arena are little to none. While they are pretty strict on their food regulations, research indicates that most of their systems are conventional rather than digital [65]. There are, however, startups and private companies that focus on enhancing digital agri-food traceability. Poland does not have its own well-established digitized ways of tracking food in the supply chain but rather adopts the EU’s main food traceability system (TRACES). The only place where an electronic system can be seen is the change in the country from paper-based agri-food customs clearance to an electronic certificate. This eases trade and also enhances food safety by having an electronic database system for trade instead.

### 4.25. Portugal

Portugal’s path to digital traceability can be outlined by the adaptation of blockchain technology in its biggest supermarket, Auchan. Auchan implemented TE-Food’s traceability system [66]. This voluntary collaboration with them began with a digital traceability system for its fresh salad, which will then be implemented on other agri-food products, too. This initiative was sparked in 2018, and it improved the ability for consumers to track their products in the country [66]. While the rest of the mandates are mostly what the EU has in place, Portugal has indicated a broader shift in the agri-food and trading industries toward a more digital approach to foster enhanced sustainability [67]. Apart from private Portuguese industries taking initiatives, the government strictly follows the EU guidelines.

### 4.26. Slovak Republic

The use of technology to mandate food chains in the Slovak Republic is an evolving field. This country also has its own national framework and policies regulated within what the EU entails. The country has, however, been proactive in terms of transparency by incorporating blockchain technologies in some aspects of its food supply chain [68]. The TE-Food blockchain system, which is the world’s largest publicly accessible solution to traceability, was supported by the Slovakian government [68]. It introduced the system of tracking the origin of food within the entire Slovak retail sector, including hotels and cafes, too. While the in-depth implementation strategies and plans are not known, the country did decide to partner up with a big blockchain company to introduce digitization in the supply chain [69].

### 4.27. Spain

In Spain, food production accounts for a substantial portion of the economy, and hence, food safety becomes a significant issue. The digital transformation in the agri-food sector in Spain is driven by the surge in Food Tech investments [70]. The Spanish government dedicated around EUR 1 million to enhance digitalization, competitiveness, and traceability. This was done via the approval of the strategic project for economic recovery and transformation (PERTE). The biggest example of digital traceability in Spain’s supply chain is the use of blockchain technology. In 2018, Carrefour launched its platform for traceability, which became the first of its kind in the country [70]. Then, by the end of 2021, Navidul, Spain’s well-known ham company, utilized blockchain for ham shoulders specifically, and it involved all consumers accessing the product’s information. Another company utilizing this was Signeblock, which basically covered the tracking of cold meats. It also guarantees the denomination of origin and the source of the raw materials used in their chorizo with IGP certification [71]. The IGP Sierra de Guadarrama is basically a certification established by the Spanish government to measure food quality standards, especially for meat. This funding and commitment to improving transparency really make Spain stand out from the other European countries [72].

Moreover, the government also promotes research and development on this narrative a lot and has funded a number of startup companies that have developed innovative systems to do so. For instance, the company Digitanimal has developed an intelligence scale for the livestock industry that allows one to keep track of the animal health status. This ensures more digital food safety. Another company, Nulab, in partnership with the country’s National Centre for Food Technology and Safety, has developed a sensory technology that measures real-time food quality and safety. This also includes hyperspectral technology, which classifies raw materials for determining the quality parameters of agri-food products [72]. Other startups include Mercatrace and Trazable, both of which are companies that offer traceability from the source to the table using a blockchain technology-backed platform called “Food Track” [70]. These startups enhance digital food safety domestically, ultimately leading to sustainable food production in Spain’s global food chain.

### 4.28. Sweden

Apart from the EU guidelines, Sweden has introduced a national food strategy that aims to promote food safety, transparency, and traceability. Its vision for 2030 revolves around sustainable and innovative food chain policies that enable growth [73]. In February 2016, the government directly mandated the Swedish National Food Agency to promote digital innovation by making information in the food supply chain easily accessible [74]. The government has set the milestone by introducing the concept of digital “Product Passports” in its plan to achieve a circular economy. Product passports will serve as tools to tell the consumer about the life cycle of the product [75]. While the EU will officially pass this regulation soon, Sween is at the forefront of testing it out and coming up with novel solutions for digital traceability. In the agri-food retail market, the classic electronic barcodes are being used to identify a product in its supply chain, but 2D codes will be implemented in the systems by 2027, which will serve as both traditional scanning at the checkout in retail as well as consumers being able to gain information about it via their smartphones. The country has initiated its path toward digitization and does have a lot of potential to do more than just follow the EU guidelines.

### 4.29. Switzerland

Switzerland is very much at the forefront of incorporating digital traceability in the agri-food industry. The fTrace solution by GS1 Switzerland is a prime example of this. fTrace is an innovative traceability solution that answers questions about a food product’s origin, processing, and quality. One of the key features of this is its reliance on the GS1 global standards [76]. It uses the Global Location Number, which is used as its unique identification number in the international supply chain. The global trade number, on the other hand, is used to ensure the consistency and reliability of the data. Moreover, fTrace has advantages in using the Electronic Product Code Information Services (EPCIS) to record every event along the supply chain of raw materials. Every processing step of the product is stored in the cloud-based fTrace database, which reduces the time-consuming process of data collection and maintenance by retailers [76]. This solution also allows consumers to gain insight into the product by having a DataBar on the product that can be scanned by those who get the smartphone app. Moreover, Switzerland, like other countries, has partnered with the food traceability firm TE-FOOD [77]. The particular partnership happened between TE-FOOD and Migros, which is Switzerland’s largest supermarket. The plan is to store the information about the products in TE’s Hyperledger-based blockchain database. This would be shared across all Migros’ outlets so that it leads to a more sustainable food supply chain. Overall, Switzerland has already established digital databases to monitor agri-food products voluntarily, apart from the guidelines that have been set up by the EU [77].

### 4.30. United Kingdom

Digital traceability has been facilitated by a range of projects and initiatives in the United Kingdom. One such project is the Digital Sandwich Project. It was launched in 2020 and was given the award of GBP 147 million for the winning applications in the UKRI Manufacturing Challenge [78]. This project aims to provide end-to-end visibility of the supply chain for sandwich ingredients [79]. It is a major collaboration between multiple stakeholders, including academics, industrialists, and government bodies, and it truly resonates with the UK’s commitment toward digital traceability [78]. It, along with many other countries, utilizes blockchain technology as well as integrated DNA technology to address a number of challenges, including consumer safety and international supply chains [79]. The Food Standards Agency has also recognized the potential of blockchain for improving food traceability and has also invested in it to provide more transparency. It took part in three trial blockchain projects that were funded by the government. They involved the beef supply chain, wine import regulation, and pork traceability [80]. In fact, with the formation of consortia like SecQual in recent years, the use of smart labels and digital IDs for products has really enhanced the farm-to-fork mindset. To illustrate the usefulness of this technology, SecQuAL’s inaugural project centered on the pig-producing sector. Here, smart labels may keep an eye on things like temperature to make sure pork products are kept at the right temperature to keep them from spoiling. Additionally, these data can be utilized to forecast the product’s shelf life, giving consumers and retailers useful information. The UK’s trade also involves a lot of digital components, which make traceability more effective [78]. The UK has passed a number of regulations with regard to import and export activities, and one of them includes the use of the Defra Digital Assistance Scheme, which allows traders and third-party software developers to share export health certificates for livestock electronically [78].

### 4.31. United States

The United States Food and Drug Administration has taken a significant step toward digitizing the food supply chain. The FDA launched this 10-year plan in July 2020 to improve food safety [81]. Therefore, it has embarked on the journey toward a ‘New Era of Smarter Food Safety’. Until recently, the records involved in moving food through the supply chain were largely paper-based, and they still are. However, this initiative wants to make sure that the digital food safety culture transcends borders. The Food Traceability Rule was passed by the government to mandate the FDA Food Safety Modernization Act [82]. It requires the manufacturing, processing, packing, or holding of foods that are on the National Food Traceability List [83]. The FSMA rule basically outlines the commodities that need to be recorded and also the ones that are at high risk of going through additional traceability regulations [84]. These records are stored in the national electronic database. When it comes to imported food products, they are subject to the same FSMA regulations and requirements as domestically produced foods. Importers need to verify the US food safety standards [85]. While regulations like the FSMA and the Food Traceability Rule provide a solid framework for implementing digital agri-food traceability, there are a lot of companies that have adopted voluntary practices. These are mostly industry-driven initiatives and consumer demand-based commitments to transparency and food safety. Examples are the produce traceability initiative and various mega companies like Costco, Wegmans, and even Walmart [84]. It can also be seen how the total investments in agri-food tech companies worldwide have grown from just over USD 2 billion in 2012 to USD 17 billion in 2018, with about half of the investments in 2018 being around USD 8 billion made by the US. This shows their commitment to not only incorporating digital traceability in their own country but also encouraging other nations to adopt it [85].

Table 2 summarizes the measures adopted by various countries to improve digital traceability in the agri-food industry. While each country has unique initiatives, blockchain technology emerges as a common theme across several nations, facilitating real-time tracking, auditing, and transparency of food products. QR codes are also widely utilized as digital identifiers, enabling consumers to access product information. Additionally, countries like Denmark, Finland, and the United Kingdom are integrating advanced technologies such as AI, IoT, and DNA technology to enhance traceability and ensure food safety. Despite varying approaches, the overarching goal is to improve transparency, safety, and sustainability throughout the food supply chain.

## 5. Discussion

### 5.1. Overview

The integration of digital traceability in the agri-food business is undergoing a substantial shift across the globe, with variable degrees of advancement and adoption. Countries are at varying levels of integrating digital traceability, and this disparity has created an intriguing landscape that provides significant insights into the future of the agri-food business. The OECD member states have initiated numerous plans, but in many cases, there have not been updated plans available that outline the current status. This was exemplified by the lack of updates from the Arla Milk Chain project in Finland, the Icelandic blockchain project with lamb products, the BrandMarks blockchain solution in Israel, and the Melons Australia and Fresh Chain collaboration, to name a few. Moreover, different countries had different levels of progression due to the expenses they could bear and how much digital traceability was prioritized. Switzerland, for example, has made tremendous progress in introducing digital traceability into its agri-food sector thanks to innovative solutions such as the fTrace platform. This system not only corresponds to worldwide standards but also makes use of cutting-edge technology to deliver an efficient and transparent supply chain tracking system.

However, it is critical to note that the implementation and adoption of such systems are dependent on a number of factors, including the cooperation of various supply chain stakeholders, the seamless integration of various technologies, and consumer willingness to engage with these new tools. Similarly, the United Kingdom has made considerable progress in its journey toward comprehensive digital traceability, as evidenced by initiatives like the Digital Sandwich Project. This project represents a significant collaboration between multiple stakeholders, including government bodies, industry players, and academic institutions, which is vital for the success of such initiatives. However, the challenge lies in ensuring the continuity and sustainability of these projects, as well as addressing any regulatory and logistical hurdles that may arise during implementation. This is in contrast to members classified as lower-to-middle-income countries who have not made significant progress in adopting digital traceability due to deeper circumstances. For instance, the lack of network coverage in rural areas brings logistical issues associated with digital extension to farms for traceability in the initial stages of supply chains. Furthermore, despite recent development success in Costa Rica, no information was found on digital traceability in the agri-food industry.

### 5.2. Collaboration

Amongst the assessed OECD countries, there was a common trend of partnerships amongst industrial producers, retailers, public and private associations, and technology providers. Together, these organizations commonly collaborated on the adoption of digital traceability. Retail and technology provider collaborations were commonly seen, especially in large European countries, such as the Auchan and Migros partnerships with Te-Foods in France and Switzerland, respectively. Other collaborations involved partnerships between producers and technology providers, such as the Barilla and IBM collaboration in Italy and the Blackbird Ventures and Emurgo partnership in Israel. Both types of collaboration hold a high value, as expertise and resources are being maximized, making for strong partnerships and streamlined adoption of digital traceability. Industry collaborations also promote the standardization of traceability systems, for instance, with the extensive use of blockchain technology across OECD countries. In terms of public collaboration, government-funded industrial initiatives were limited, as most agri-food startups launching digital traceability initiatives were private. Notably, many of the government-driven initiatives in member countries were independent of industry companies.

### 5.3. Government vs. Industry Approaches

Given the contrast in government-led initiatives when compared to industry-based initiatives, one common theme that arose is the difference in approaches behind the adoption of digital initiatives. Government-led initiatives were most often seen as a response to heightened food safety concerns or a crisis, as seen in the case of Japan’s response to the “mad cow disease” outbreak or Norway’s response to the E. coli outbreak. This also leads to the focus being more on food safety and preventative measures than traceability. While food safety is critical for public health, it may not fully address transparency and traceability. Furthermore, hyperfocusing on food safety may jeopardize all-inclusive traceability systems, where a product’s end-to-end visibility in the supply chain will be limited. This is in contrast with industry-led initiatives, which solely focus on building consumer trust and thus fulfilling their wants and needs. This can be seen in almost all industry-based efforts, for instance, with Aeon in Japan and Carrefour in Spain having integrated technology to build consumer confidence. As more and more consumers demand full transparency in food products, the future of industry-led initiatives will likely revolve around the greater integration of leading technologies such as blockchain as well as continuous improvement in such technologies. This would allow consumers to see greater detail in the presentation of food products as they move through the supply chain.

### 5.4. Digital Technology

Given that the use of blockchain technology for digital traceability is a recurring theme in several countries, it will be a cornerstone for digitization in the agri-food industry. One of the primary reasons why this is the case is the high level of data security that supplements traceability and transparency. Consequently, blockchain has been deemed a versatile and trusted solution for not only food traceability but also food safety and, thus, consumer trust as well. This was exemplified by the fact that most countries used it for high-value and high-trust products such as meat, dairy, and seafood. In most cases, the most commonly used digital identifiers were QR codes. Often supplemented with blockchain, QR codes provide one of the most user-friendly solutions to digital traceability, allowing for high accessibility throughout the entire supply chain. Collectively, their high level of effectiveness and efficiency in traceability attribute them to helping shape future digital traceability initiatives.

In addition to the benefits of technology in traceability discussed so far, governments can play a crucial role in establishing regulatory frameworks and standards for the implementation of technology (i.e., blockchain-based traceability systems) in the agri-food industry. This includes creating policies that promote the adoption of blockchain and/or other technologies, providing incentives for companies to invest in digital traceability solutions, and ensuring compliance with data security and privacy regulations [86]. Furthermore, governments can collaborate with industry stakeholders to develop comprehensive strategies for integrating technology into existing supply chain operations. This involves identifying key areas for improvement, such as enhancing transparency, reducing food fraud, and improving the efficiency of recalls and crisis management. By actively participating in the development and implementation of blockchain-based traceability systems, governments can demonstrate their commitment to ensuring food safety and consumer trust in the agri-food sector [87].

Private companies operating in the agri-food supply chains also have important managerial considerations to address. This includes conducting cost-benefit analyses, evaluating the scalability of blockchain solutions and other technologies, and assessing the impact on existing supply chain processes [88]. Moreover, private companies must invest in employee training and capacity-building to ensure the smooth implementation and adoption of digital traceability systems [89]. Overall, governments and private companies need to collaborate closely to leverage the full potential of technology in enhancing digital traceability in the agri-food industry. By addressing managerial considerations such as regulatory compliance, strategic planning, and technological integration, stakeholders can effectively harness the benefits of blockchain or other important technologies to improve food safety, traceability, and consumer trust.

## 6. Limitations

Several limitations, which have implications for the interpretation and application of our findings, were encountered while performing this research. These constraints are essential for comprehending the complexities of the OECD countries’ digital traceability systems. Firstly, there is a noticeable dearth of comprehensive information on digital traceability. The sources available were quite limited, and the government websites did not have the frameworks surrounding the digital component of traceability. They were more extensive on the regular traceability frameworks, and a lot of the countries in the OECD still heavily rely on paper-based traceability. Moreover, some resources available were only in the language of the member state; hence, the barrier to gaining the information from that also results in the omission of certain data and valuable insights to the countries’ analysis. Furthermore, the rapid evolution of digital traceability technology also makes the whole process of gathering data complex. It is also because of the novel nature of the involvement of technology in the agri-food sector. Notably, farmers in Canada and Colombia had low adherence to digital traceability due to the complexities associated with it, as well as preferred manual practices. Consumers are also more aware of sustainability now, and in order to meet their demands, certain measures have to be taken. This changing dynamic is accompanied by changing technologies; hence, data on such futuristic matters gets tedious to gather. Last but not least, the costs and infrastructure associated with it are quite expensive. Specifically, in low-resource communities like Colombia, it becomes difficult to implement such novel systems. Even countries like Mexico, Costa Rica, and Turkey did not have data on them; part of the reason was the lack of awareness and infrastructure. If all these limitations were overcome, a more holistic and nuanced understanding of the regulated and voluntary digital traceability framework across the OECD member states would be possible.

## 7. Future Directions

The current initiatives taken in the agri-food sector can be further improved and made more extensive across all the OECD member states. There is still a dire need for this digitalization to be applied to more commodities, as the current ones only apply to certain high-value products, usually livestock or certain fruits and vegetables. This expansion would make a country excel in the fields of digital traceability and consumer food safety. Moreover, currently, all the digital initiatives are mostly from private startup companies and do not entail legal government regulations that are mandatory for the country to follow. Furthermore, these digital traceability systems must be considered essential components of the larger agricultural ecosystem rather than separate entities. For instance, given the differences in government-led initiatives when compared to industry-led initiatives, it would be ideal to integrate both approaches. This would entail government organizations strengthening food safety regulations while also integrating advanced technologies and innovating. This holistic approach to combining government regulations and industry-led innovations would facilitate a safer and more transparent food supply chain. Lastly, every initiative would benefit from a continuous monitoring system of implemented initiatives, as many of the digital traceability practices discussed in this paper recently began and, therefore, could not be assessed for their efficacy. This would allow for the improvement of existing technologies while continuing to explore innovative, advanced software. Overall, more incorporation of digitization in the agri-food sector needs to be implemented so that consumers are more aware and the issues that arise from the lack of traceability, like foodborne illnesses, are reduced.

## 8. Conclusions

As we embrace the complexities of the Industry 4.0 era, the significance of digitization in the agri-food industry becomes increasingly pronounced, particularly in ensuring food safety and transparency for consumers. This comprehensive review paper delved into the nuanced landscape of digital traceability across all 38 OECD member countries, meticulously evaluating each nation’s adoption and consequences of digital traceability within their agri-food supply chains. By using a comprehensive analytical framework, this study highlighted the diverse approaches and varying degrees of progressiveness exhibited by participating countries, underscored by a notable dearth of updated information on ongoing initiatives. Notably, the analysis identified prevalent patterns of high-value industrial collaborations alongside discernible disparities in the strategies pursued by private and public entities. Furthermore, the ubiquitous adoption of blockchain technology and QR codes emerged as prominent trends across all member countries, underscoring their pivotal role in enhancing traceability and transparency. Despite the manifold benefits of digital traceability in the sector, this study also elucidated significant challenges faced by the sector, prompting a nuanced discussion on future directions and potential solutions. In all, the evolution of digital traceability represents a transformative journey toward cultivating a safer, more traceable, and transparent future for the agri-food industry, underpinned by ongoing advancements in digital technology and collaborative efforts across diverse stakeholders. Moving forward, continued research and innovation in this field will be essential to address emerging challenges and drive sustainable growth in the agri-food sector.

The comparative analysis of digital traceability in agri-food supply chains across OECD member countries has underscored several strengths and areas for improvement. One of the principal strengths is the capacity of digital traceability systems to enhance food safety and supply chain transparency. These systems facilitate real-time tracking of food products, enabling swift responses to food safety incidents and improving consumer trust. Moreover, the integration of technologies like blockchain and IoT has shown potential in ensuring the authenticity and integrity of agri-food products, aligning with global sustainability goals.

However, the study also revealed areas for improvement in the deployment of digital traceability systems. These include the need for standardized regulations and frameworks across countries to ensure seamless integration and interoperability of traceability systems. Additionally, there is a pressing requirement for capacity-building initiatives to educate stakeholders, including farmers, producers, and supply chain managers, on the benefits and operational aspects of digital traceability.

The study acknowledges several limitations that could impact the generalizability and applicability of its findings. Firstly, the rapid evolution of digital technologies and their application in agri-food traceability presents a challenge in maintaining up-to-date and comprehensive data across all OECD member countries. Secondly, the diversity in economic, technological, and regulatory environments across these countries complicates the development of a one-size-fits-all approach to digital traceability. Lastly, the study’s focus on OECD member countries means that the findings may not fully represent the global state of digital traceability in agri-food supply chains, especially in developing and non-OECD economies.

While digital traceability in agri-food supply chains offers significant benefits in terms of enhancing food safety and operational efficiency, it also necessitates a concerted effort to address the existing challenges. Future research should aim to develop inclusive frameworks that can accommodate the diverse needs and capabilities of different countries, thereby fostering a more integrated and efficient global agri-food traceability system.

## Figures and Tables

**Figure 1 foods-13-01075-f001:**
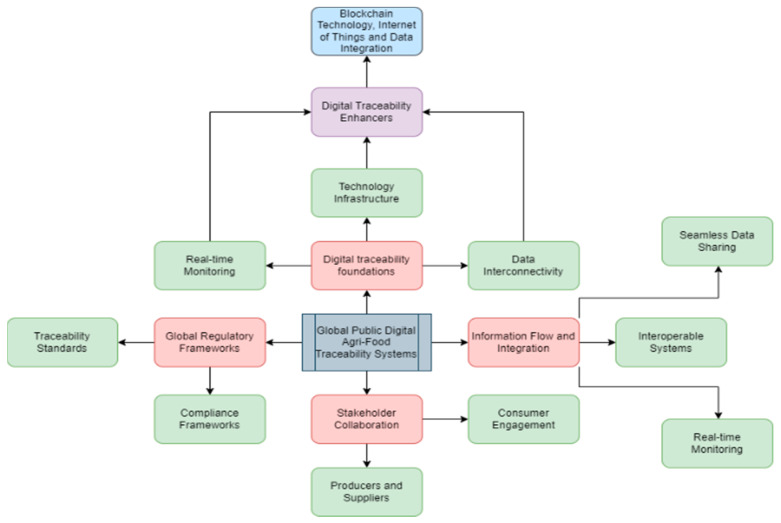
Global public digital agri-food traceability systems.

**Table 1 foods-13-01075-t001:** Assessment Questions.

Sl.	Questions
1.	Are there specific and well-defined digital traceability regulations or policies established at the national or regional level?
2.	What are the guidelines and practices regarding digital traceability when it comes to imported products? What are the practices involved in that?
3.	To what extent is the regulatory authority responsible for overseeing digital traceability regulations?
4.	In instances where there are no mandatory regulations, are there voluntary digital traceability practices or industry-driven initiatives that have been adopted?
5	Within the agri-food supply chain of the country, which specific products or commodities are subject to digital traceability regulations and monitoring?
(a)	What forms of digital identifiers or codes are employed for tracking and registering imported products within the digital traceability system? This may encompass digital barcodes, RFID codes, or other such identifiers.
(b)	Does the country maintain an electronic database system dedicated to monitoring imports and exports, inclusive of their respective digital traceability data?
6.	What specific digital traceability information is provided on product packaging labels to empower consumers with insights into the product’s origin and its journey within the supply chain?

**Table 2 foods-13-01075-t002:** Summary of country-level initiatives for digital traceability.

Countries	Summary of Measures Adopted to Improve Digital Traceability
Australia	RegTech: Main technology for digital traceability. Agrifood Connect Trace2Place: Real-time mapping of the red meat supply chain. Digital exchange systems: Streamline data collection for risk assessment. eCert: Electronic certification system for tracking imports and exports.
Belgium	QR codes are utilized for food traceability. Adoption of EU-led initiatives and policies.
Canada	Limited adoption of digital traceability. Participation in the GS1 global system of standards for digital safety. Industry-led initiative: The Consortium of Parmigiano Reggiano partners for digital identifiers.
Chile	Integrated traceability solution for Chilean seafood produce by Shellcatch. Utilization of vessel, coastal, and scale cameras feeding data to a Cloud-based eMonitoring system. AI-powered data collection, analysis, and management. QR codes on products serve as digital identifiers.
Colombia	Limited network coverage in rural areas hinders digital extension for traceability. Government-led initiatives: Coffee Information System (SICA). Emerging digital startups like Curuba Tech and platforms like Trusty.
Denmark	Denmark lags in agri-food startup investment despite being highly digitalized.
Finland	Blockchain technology for supply chain transparency. Implementation of AI for real-time updates on animal welfare. Integration of IoT to automate blockchain technology for traceability.
France	blockchain technology for real-time tracking and auditing of food products. QR codes for product history.
Germany	SiLKe project on blockchain technology for a safe food chain.
Greece	Greece lacks specialized initiatives or guidelines for digital traceability
Hungary	Hungary lacks specific initiatives for digitalizing food systems but plans to adopt digital technologies in the agricultural sector, aligning with EU guidelines. Top of Form
Iceland	Blockchain-based lamb traceability, but no updates since, and reliance on non-digital food safety regulations. Top of Form
Ireland	AI applications and ICT for traceability in food. Part of Ireland’s Food Vision 2030 strategy is embracing the digital revolution for improved food system transparency.
Israel	Blockchain to provide consumers with detailed food product information, although no updates have been provided since 2020.
Italy	Blockchain for certifying food product information. IoT platform with IT service providers for digital traceability.
Japan	Individual Identification Register system after the “mad cow disease” outbreak challenges in fully digitizing traceability due to limited pilot project adoption and budget constraints.
Republic of Korea	blockchain technology QR code-based verification systems.
Latvia	QR Codes Mobile Applications with a traceability system in Latvia, Fisheries Integrated Control and Information System by the Latvian Ministry of Agriculture.
Lithuania	Artificial intelligence into digital traceability solutions. Distributional blockchain systems integrated into the AgriFood sector.
Netherlands	e-CertNL System facilitating the issuance of export and health certificates for agricultural products. Blockchain Technology QR Codes
New Zealand	QR Codes IDlocate: electronic database tool generating unique URLs for each food product. Blockchain Technology NAIT Scheme: tag animals and record their identities in the national database.
Norway	eSporing Traceability Project: focusing on establishing electronic infrastructure for information exchange in the food supply chain. Radio Frequency Identification (RFID) Technology: with GS1 electronic product code standards. Blockchain Collaboration in the Seafood Sector
Poland	EU Regulations Compliance minimal independent initiatives in this area, with most systems remaining conventional rather than digitized.
Portugal	Blockchain in Auchan Supermarket, TE-Food’s traceability system. a broader shift toward digital approaches in the agri-food and trading industry. Adherence to EU Guidelines
Slovak Republic	Blockchain in Food Supply Chain Tracking Origin of Food within the entire Slovak retail sector.
Spain	Blockchain Technology
Sweden	Digital “Product Passports” to inform consumers about the life cycle of products. Barcode Evolution for both traditional checkout scanning and consumer access to product information via smartphones.
Switzerland	fTrace Solution by GS1, which utilizes the Electronic Product Code Information Services (EPCIS) to record events along the supply chain and store processing steps in a cloud-based database. TE’s Hyperledger-based blockchain database enhances the sustainability of the food supply chain.
United Kingdom	Blockchain and DNA Technology to address challenges, including consumer safety and international supply chains. SecQual Consortium to enhance traceability with smart labels and digital IDs for products.
United States	FDA’s 10-year plan launched in July 2020 under the “New Era of Smarter Food Safety” initiative. Food Traceability Rule, requiring records for manufacturing, processing, packing, or holding foods to be stored in a national electronic database. Imported Food Products Regulation, requiring imported food products to comply with FSMA regulations. Industry-Driven Initiatives.

## Data Availability

The original contributions presented in the study are included in the article, further inquiries can be directed to the corresponding author.

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
