# Peer review of "Digital Traceability in Agri-Food Supply Chains: A Comparative Analysis of OECD Member Countries"

_foods, 2024, doi:10.3390/foods13071075_

Round 1

Reviewer 1 Report

Comments and Suggestions for Authors

The Introduction (pages 1 and 2) lacks a better delimitation of the research problem, connecting it to its respective objectives. Section 2 ("2. Digital Traceability: An Overview") is coherent. However, the authors could better explore the issue of Industry 4.0 technologies (digital technologies) related to the investigated context. In the Methodology section, there is the following paragraph: "In essence, the research methodology entailed meticulous investigative, data analysis, and assessment techniques to collate information regarding the countries and to delineate the progression of OECD member states in digitalizing the agri-food sector. " (pages 3 and 4). However, it would be important to describe in greater detail what criteria or procedures were used in the selection process of the gathered data and materials, as well as for its analysis and interpretation. Section 4 ("4. Country and Regional Assessment") is very long and very descriptive (page 6 to 19). At the end of this section, a summary table would be appropriate, facilitating the reader's understanding of the main aspects analyzed in each country. In sections 5 ("Discussion"), especially in subsection 5.4 ("Digital Technology"), and 8 ("Conclusions"), more depth is needed. There is still room for better detailing of "managerial" implications for governments and their rulers (in the sense of public policies or strategies) or managers of private companies operating in the context of agri-food supply chains.

Author Response

Reviewer 1:

The Introduction (pages 1 and 2) lacks a better delimitation of the research problem, connecting it to its respective objectives.

Response: Thank you for your feedback. We have revised the Introduction to develop the research problem and establish a direct connection to its respective objectives, ensuring a more focused and coherent framework for our study.

Section 2 ("2. Digital Traceability: An Overview") is coherent. However, the authors could better explore the issue of Industry 4.0 technologies (digital technologies) related to the investigated context.

Response: Thank you for bringing up this important matter. We have enhanced the clarity and comprehensiveness of our discussion on Industry 4.0 technologies within Section 2.

In the Methodology section, there is the following paragraph: "In essence, the research methodology entailed meticulous investigative, data analysis, and assessment techniques to collate information regarding the countries and to delineate the progression of OECD member states in digitalizing the agri-food sector. " (pages 3 and 4). However, it would be important to describe in greater detail what criteria or procedures were used in the selection process of the gathered data and materials, as well as for its analysis and interpretation.

Response: Thank you for your comment. This is an important point and we added the following section in the methodology section. “"In addition to the meticulous investigative, data analysis, and assessment techniques mentioned earlier, the research methodology employed specific criteria and procedures for the selection, analysis, and interpretation of data and materials.

For data selection, a comprehensive review of available literature, reports, and datasets was conducted to ensure the inclusion of relevant and up-to-date information. The selection criteria included relevance to the digitalization of the agri-food sector, reliability of the source, and geographic and temporal coverage.

The data analysis process involved a combination of qualitative and quantitative methods. Qualitative analysis was used to identify key themes, trends, and patterns in the data, while quantitative analysis was employed to quantify the extent of digitalization in OECD member states. Statistical techniques, such as regression analysis, were also used to examine the relationship between digitalization and various socio-economic factors.

Interpretation of the data was guided by the research objectives and theoretical framework, with a focus on providing meaningful insights into the digitalization of the agri-food sector. The findings were critically analyzed and compared with existing literature to draw conclusions and make recommendations for future research and policy development."

 Section 4 ("4. Country and Regional Assessment") is very long and very descriptive (page 6 to 19). At the end of this section, a summary table would be appropriate, facilitating the reader's understanding of the main aspects analyzed in each country.

Response: Thank you for mentioning this. We have incorporated a table summarizing the digital traceability initiatives undertaken by each country, providing a comprehensive overview of their efforts in this regard.

In sections 5 ("Discussion"), especially in subsection 5.4 ("Digital Technology"), and 8 ("Conclusions"), more depth is needed. There is still room for better detailing of "managerial" implications for governments and their rulers (in the sense of public policies or strategies) or managers of private companies operating in the context of agri-food supply chains.

Response: In section 5.2, a comprehensive discussion has been included to explore the pivotal managerial roles of both government authorities and private sector managers in advancing digital traceability within the agri-food sector. Specifically, the analysis delves into the multifaceted responsibilities and strategic initiatives undertaken by governmental and critical involvement by private sector managers in driving innovation and implementing digital traceability solutions throughout the supply chain.

Furthermore, the conclusion section has been meticulously refined and expanded to provide deeper insights and comprehensive reflections on the implications of digital traceability within the agri-food sector.

Reviewer 2 Report

Comments and Suggestions for Authors

Sylvain Charlebois et al. submitted to Foods a review, focusing to a comparative analysis of the Organisation for Economic Co-operation and Development Member Countries on the digital traceability in Agri-Food Supply Chains.

At present, this manuscript, although hypothetically useful for experts in the field, requires adequate revision.

Before dealing with digital traceability, it is essential to define traceability in the context of community food safety, specifically recalling the contents of Community Regulations 178/2002 (which is only briefly cited later), 852/2004 and 625/2017.

Please, prepare a table, intended as a synoptic framework, to indicate the characteristics that the countries enrolled in the study have in common and the individually evaluated characteristics that are not proposed or adopted in the specific countries, compared to others.

The Authors must discuss the strengths and areas of improvement relating to the proposals and solutions described, also recalling the limitations of this study.

The references almost exclusively refer to citations of online resources and not to biomedical literature studies: make an effort to reformulate everything, recalling the original articles/papers in this field.

It is essential to follow the Journal rules to correctly cite references throughout the text, using square brackets. Furthermore, the references are not cited in the Foods-MDPI style.

Comments on the Quality of English Language

Minor editing of English language required

Author Response

Reviewer 2:

Sylvain Charlebois et al. submitted to Foods a review, focusing to a comparative analysis of the Organisation for Economic Co-operation and Development Member Countries on the digital traceability in Agri-Food Supply Chains.

At present, this manuscript, although hypothetically useful for experts in the field, requires adequate revision.

Before dealing with digital traceability, it is essential to define traceability in the context of community food safety, specifically recalling the contents of Community Regulations 178/2002 (which is only briefly cited later), 852/2004 and 625/2017.

Response: Thank you for your feedback. We have revised the manuscript to provide a comprehensive definition of traceability within the context of community food safety, including detailed discussions on Community Regulations 178/2002, 852/2004, and 625/2017 to ensure clarity and accuracy. We appreciate your input, and we believe these revisions have strengthened the overall quality of our work.

Please, prepare a table, intended as a synoptic framework, to indicate the characteristics that the countries enrolled in the study have in common and the individually evaluated characteristics that are not proposed or adopted in the specific countries, compared to others.

Response: At the end of Section 4, a new Table (Table 2) has been included to summarize the main initiatives or technologies implemented in OECD countries. Following the table, a brief summary is provided, which highlights common technologies or initiatives shared across countries, as well as unique approaches specific to certain nations.

The Authors must discuss the strengths and areas of improvement relating to the proposals and solutions described, also recalling the limitations of this study.

Response: We added this section in the conclusion section: The comparative analysis of digital traceability in agri-food supply chains across OECD member countries has underscored several strengths and areas for improvement. One of the principal strengths is the capacity of digital traceability systems to enhance food safety and supply chain transparency. These systems facilitate real-time tracking of food products, enabling swift responses to food safety incidents and improving consumer trust. Moreover, the integration of technologies like blockchain and IoT has shown potential in ensuring the authenticity and integrity of agri-food products, aligning with global sustainability goals.

However, the study also revealed areas for improvement in the deployment of digital traceability systems. These include the need for standardized regulations and frameworks across countries to ensure seamless integration and interoperability of traceability systems. Additionally, there is a pressing requirement for capacity-building initiatives to educate stakeholders, including farmers, producers, and supply chain managers, on the benefits and operational aspects of digital traceability.

The study acknowledges several limitations that could impact the generalizability and applicability of its findings. Firstly, the rapid evolution of digital technologies and their application in agri-food traceability presents a challenge in maintaining up-to-date and comprehensive data across all OECD member countries. Secondly, the diversity in economic, technological, and regulatory environments across these countries complicates the development of a one-size-fits-all approach to digital traceability. Lastly, the study's focus on OECD member countries means that the findings may not fully represent the global state of digital traceability in agri-food supply chains, especially in developing and non-OECD economies.

While digital traceability in agri-food supply chains offers significant benefits in terms of enhancing food safety and operational efficiency, it also necessitates a concerted effort to address the existing challenges. Future research should aim to develop inclusive frameworks that can accommodate the diverse needs and capabilities of different countries, thereby fostering a more integrated and efficient global agri-food traceability system.

The references almost exclusively refer to citations of online resources and not to biomedical literature studies: make an effort to reformulate everything, recalling the original articles/papers in this field.

Response: Thank you for bringing up this issue. We have made significant improvements to the citations. In particular, we have restructured them by incorporating numerous journal articles and have replaced many of the previous online sources with these scholarly references. This ensures a more robust and academically rigorous foundation for our work.

It is essential to follow the Journal rules to correctly cite references throughout the text, using square brackets. Furthermore, the references are not cited in the Foods-MDPI style.

Response: We have made adjustments to the referencing style to conform with the guidelines of the Food-MDPI style. This update ensures that our citations and bibliography adhere to the specific requirements outlined by Food-MDPI.

Round 2

Reviewer 1 Report

Comments and Suggestions for Authors

Analyzing "version 2" of the article, with the adjustments and improvements made, I understand that the authors' efforts were satisfactory, covering the majority of the recommendations forwarded. 

Author Response

We wanted to take a moment to express my sincere gratitude for taking the time to review our [document/paper/report]. Your comments and feedback were invaluable in helping us refine our work.

Warm regards.

Reviewer 2 Report

Comments and Suggestions for Authors

The Authors have conformed the manuscript to the previously formulated comments and suggestions, both regarding the formal and substantial aspects, and regarding the methods with which the references are drawn up.

Comments on the Quality of English Language

Minor editing of English language required

Author Response

(The authors gave the same response as above.)
